# A Comparative Assessment of Sentinel-2 and UAV-Based Imagery for Soil Organic Carbon Estimations Using Machine Learning Models

**DOI:** 10.3390/s25175281

**Published:** 2025-08-25

**Authors:** Imad El-Jamaoui, Maria José Martínez Sánchez, Carmen Pérez Sirvent, Salvadora Martínez López

**Affiliations:** Department of Agricultural Chemistry, Geology and Pedology, Faculty of Chemistry, University of Murcia, 30100 Murcia, Spain; mjose@um.es (M.J.M.S.); melita@um.es (C.P.S.)

**Keywords:** UAV imagery, Sentinel-2 satellite imagery, machine learning, soil organic carbon (SOC), laboratory analysis, remote sensing, precision agriculture

## Abstract

As the largest carbon reservoir in terrestrial ecosystems, soil organic carbon (SOC) plays a critical role in the global carbon cycle and climate change mitigation. A promising approach to swiftly procuring geographically dispersed SOC data is the amalgamation of UAV-based multispectral imagery at the local scale and Sentinel-2 satellite imagery at the regional scale. This integrated approach is particularly well-suited for precision agriculture and real-time monitoring. In this study, we evaluated the performance of UAVs and Sentinel-2 imagery in predicting SOC using four machine-learning models: Multiple Linear Regression (MLR), Support Vector Regression (SVR), Random Forest (RF), and Artificial Neural Networks (ANNs). UAV imagery outperformed Sentinel-2, achieving more accurate detection of local SOC variability thanks to its finer spatial resolution (5–10 cm versus 10–20 m). Among the models tested, the Random Forest algorithm achieved the highest accuracy, with an R^2^ of up to 0.85 using UAV data and 0.65 using Sentinel-2 data, along with low RMSE values. All models confirmed the superiority of UAV imagery based on key error metrics (SSE, MSE, RMSE, and NSE). Although Sentinel-2 remains valuable for regional assessments, UAV imagery combined with Random Forest provides the most reliable SOC estimates at local scales. The spatial SOC maps generated from both UAV and Sentinel-2 imagery showed more nuanced spatial variability than standard interpolation techniques. While prediction accuracy using UAV-based models was slightly lower in some cases, UAV imagery provided greater spatial detail in SOC distribution. However, this is associated with higher acquisition and processing costs compared to freely available Sentinel-2 imagery. Given their respective advantages, we recommend using UAV imagery for detailed, site-specific SOC estimations and Sentinel-2 data for broader regional-to-global SOC mapping efforts.

## 1. Introduction

Soil represents the most significant terrestrial carbon reservoir, accounting for approximately 75% of the total carbon stock within terrestrial ecosystems. Even minor alterations in soil carbon stocks can exert a substantial influence on atmospheric CO_2_ levels and global climate dynamics [1]. Soil organic carbon (SOC) has been identified as a key indicator of soil fertility and quality, playing an important role in maintaining soil health and supporting various ecosystem functions [2,3]. Accurate estimation of SOC is imperative for comprehending soil carbon dynamics and implementing effective land management practices. However, the direct measurement of soil carbon remains challenging due to the limitations of traditional methods, which vary in terms of directness, accuracy, and assumptions [4]. Although laboratory analysis via dry combustion is widely regarded as the gold standard for SOC determination, it is costly and time-consuming for large-scale applications [5,6].

Consequently, spatial detection and mapping of soil organic carbon (SOC) in agricultural land is imperative for the purpose of monitoring soil quality and achieving environmental benefits [7,8]. It is evident that elevated levels of soil organic carbon (SOC) have a positive impact on soil structure, water retention, and nutrient availability. This, in turn, has been shown to enhance crop productivity [9]. The integration of multispectral remote sensing data from satellites, such as Sentinel-2A, has been identified as a promising method for the estimation of soil properties over extensive areas. This technique facilitates the analysis of spectral responses, thereby enabling the inference of spatial variability in soil properties [10]. Sentinel-2, a constituent element of the European Space Agency’s Copernicus programme, facilitates the acquisition of high-resolution optical imagery for the purpose of terrestrial monitoring, encompassing the analysis of soil and vegetation characteristics [11]. Vegetation indices, including NDVI, SAVI, and EVI, have emerged as favoured proxies for SOC estimation due to their non-destructive nature and wide applicability on a large scale [12]. The utilisation of spectral bands, exhibiting sensitivity to vegetation and soil conditions, by these indices facilitates the process of SOC mapping. However, it should be noted that the accuracy of such measurements can be affected by confounding factors such as soil moisture, vegetation cover, and atmospheric variability. Therefore, careful calibration is required [13].

Machine learning (ML) and artificial intelligence (AI) have witnessed significant advancements in recent times, resulting in substantial enhancements in the capabilities of systems to estimate SOC. For instance, Chen [14] utilised machine-learning (ML) algorithms, including Random Forest (RF), Extreme Gradient Boosting (XGBoost), and a Support Vector Machine (SVM), in conjunction with remote sensing data, to estimate surface soil organic carbon (SOC) in Shandong, China. The XGBoost model demonstrated the most optimal outcomes (R^2^ = 0.7548, RMSE = 7.68, RPD = 1.13), with elevation and clay content identified as the most substantial predictive variables. Moreover, a systematic review by Lima [15] emphasised the increasing utilisation of deep learning (DL), neural networks, and ensemble methods in SOC estimation, underscoring their advantages across diverse spatial and temporal scales. In a similar vein, Zhang [16] demonstrated that the incorporation of environmental covariates, such as topography, texture and climate data, into Support Vector Machine (SVM) models resulted in a substantial enhancement of soil organic carbon (SOC) predictions (R^2^ = 0.959, RMSE = 13.99). Moreover, there has been an increasing focus in academic studies on comparing the effectiveness of UAV-based and Sentinel-2 satellite imagery in terms of their ability to estimate SOC across a range of spatial scales. Unmanned aerial vehicles (UAVs) equipped with high-resolution multispectral sensors have been shown to provide centimetre-level detail, thereby enabling more precise soil organic carbon (SOC) predictions at a local level (R^2^ ≈ 0.84). This high level of precision can be attributed to the capacity of UAVs to capture fine-grained spatial variability and adapt to site-specific conditions [17,18]. Conversely, Sentinel-2 imagery is freely available, frequent, and provides extensive coverage at medium resolution (10–60 m). However, it tends to be less accurate (R^2^ ≈ 0.69) in heterogeneous environments due to spectral mixing and atmospheric disturbances [19,20].

Despite their high accuracy, UAVs are resource-intensive and less scalable for regional or national applications due to operational constraints and the processing of large amounts of data. Conversely, the Sentinel-2 satellite offers enhanced accessibility and spatial coverage, rendering it a more practical option for large-scale SOC monitoring, albeit at the expense of reduced spatial detail [21]. A combination of data sources, utilising machine-learning algorithms such as Random Forest, has been shown to yield enhanced performance (R^2^ ≈ 0.76). This enhancement is achieved through the integration of UAV-derived data for local calibration and Sentinel-2 imagery for broader generalisation [22]. Recent advancements in the field of remote sensing have indicated that unmanned aerial vehicle (UAV) imagery is highly effective for precision agriculture and fine-scale soil monitoring. In contrast, the Sentinel-2 mission continues to demonstrate its superiority in large-scale soil organic carbon (SOC) mapping. New frameworks have been developed that allow UAV-derived SOC estimates to be scaled up to Sentinel-2 imagery, which significantly improves the assessment of SOC in complex or degraded environments, such as post-mining landscapes. The results of the present study lend support to the notion that the integration of laboratory methodologies with remote sensing data and machine-learning models is a pivotal factor in optimising the precision of SOC mapping. It is evident that such approaches have the capacity to enhance efficiency and facilitate the implementation of sustainable land management strategies that are aligned with the objectives of climate change mitigation. Although previous studies have explored the use of UAV and Sentinel-2 data for estimating soil organic carbon (SOC), the novelty of the present research lies in its integrated, multi-scale approach, which combines both data sources within a unified machine-learning framework. Furthermore, the study assesses the applicability of UAV and Sentinel-2 data for SOC estimation within a semi-arid agro-ecological system in Murcia, providing insight into model scalability and performance in complex but relatively homogeneous landscapes. The methodology has been applied to a complex agro-ecological region characterised by hydrological variability and land use patterns. This research addresses a gap in existing literature and contributes new evidence to support multi-scale remote sensing strategies for SOC monitoring.

The primary objective of this study is to evaluate and compare the effectiveness of UAV-based high-resolution imagery and Sentinel-2 satellite data in estimating soil organic carbon (SOC) across cereal-based agricultural landscapes, with an emphasis on spatial accuracy and predictive performance. Specifically, the study appraises and contrasts the prediction accuracy of soil organic carbon (SOC) models derived from spectral bands and vegetation indices of unmanned aerial vehicle (UAV) and Sentinel-2 data. Furthermore, it evaluates the influence of spatial resolution and spectral characteristics on SOC estimation at local and regional scales. In addition, it analyses the individual strengths and limitations of each data source in capturing spatial SOC variability. Furthermore, the study investigates the benefits of combining UAV and Sentinel-2 datasets for multi-scale SOC mapping, with a view to enhancing spatial precision and model robustness. Machine-learning algorithms are employed to develop and validate SOC estimation models using UAV data, Sentinel-2 data, and their combination to determine the most accurate and scalable approach. Finally, the study proffers recommendations for effective remote sensing strategies for SOC monitoring, accounting for the trade-offs between resolution, spatial coverage, and predictive performance.

## 2. Materials and Methods

### 2.1. Study Area Soil Sampling and Laboratory Analysis

The study site, located in Murcia, covers an area of 30 hectares and is situated in southern Spain (Figure 1). This location plays a crucial role as an integral part of the LIFE AMDRYC4 project. In the Murcia Region, prevailing wind speeds are consistently around 6 km/h, with a humidity level of 74%. The average elevation in the area is 78 m above sea level. The landscape mainly consists of winter and spring cereals cultivation, showcasing diverse topography with valleys and slopes. The predominant soil types are Calcisols and Regosols, distributed across crystalline and sedimentary rock formations, as classified by the World Reference Base for Soil Resources. Almonds and olives are the primary crops cultivated in this region.

A total of 70 soil samples were collected within a 10-metre radius from the centre of each grid cell at depths ranging from 0 to 30 centimetres in July 2022. The sampling locations were meticulously documented using a hand-held GPS device to ensure spatial precision. The samples were subjected to air-drying, followed by grinding and sifting to a size of 2 mm. Subsequent to this, laboratory analysis was conducted. The sample size was selected to achieve a balance between field resource constraints and the necessity for sufficient data to train and validate machine-learning models for the estimation of soil organic carbon (SOC). While the dataset is adequate for assessing model performance and drawing initial conclusions at the local scale, its relatively limited size may restrict the scalability and generalizability of the models to broader or more diverse regions.

### 2.2. Laboratory Analysis

The wet oxidation technique is based on the oxidation of carbon (C) using potassium dichromate (K_2_Cr_2_O_7_) in sulphuric acid (H_2_SO_4_). This method results in incomplete oxidation of total organic compounds, as the most reactive organic carbon fractions are primarily oxidised. The Walkley–Black method for measuring soil organic carbon (SOC) was described by Nelson and Sommers. In this method, 1 g of oven-dried (40 °C), finely ground soil (sieved to 250 µm), is placed in a 125 mL conical flask. Then, 10 mL of 0.2 M potassium dichromate is added, followed by 10 mL of concentrated sulphuric acid, added slowly. In the case of soils with a higher *SOC* content (>4%), a larger volume of potassium dichromate and sulphuric acid is required to ensure sufficient oxidation. After 30 min of oxidation at room temperature, 50 mL of distilled water, 3 mL of concentrated orthophosphoric acid (H_3_PO_4_), and four drops of diphenylamine indicator are added. The excess potassium dichromate is then titrated with 0,1 M Mohr’s solution to determine the amount of *SOC* present. For each batch of soil samples, three blanks are titrated for the accurate determination of the molarity of the Mohr’s salt solution. Use the following formula to calculate the *SOC* concentration (in g C/kg soil)$$SOC(g\;C\;{kg}^{-1})=\frac{\left(\nu_{b}-v_{s}\right)\times C_{Fe^{2+}}*0.003*1000}{weight\;of\;sample\;(g)}$$
where

ν*_b_* = volume (mL) of Mohr’s salt used for the blank titration.

ν*_s_* = volume (mL) of Mohr’s salt used for the sample.

$C_{Fe^{2+}}$ = molarity of Mohr’s salt solution.

The factor 0.003 g/mmol is derived from (0.012/4), where 0.012 g/mmol is the molar mass of carbon, and 4 is the number of electrons transferred during the oxidation of one mole of carbon.

To correct for incomplete oxidation, the SOC content is multiplied by a correction factor of 1.33, as recommended by Nelson and Sommers [23]. This factor compensates for the fraction of organic carbon not fully oxidised under Walkley–Black conditions, especially in soils of variable composition or management [24].

### 2.3. UAS Multispectral Imagery Data Acquisition

Multispectral data was acquired using a Trinity F90 fixed-wing drone equipped with a MicaSense Altum dual sensor, which included two cameras: one for RGB imaging, and the other for multispectral acquisition. The multispectral sensor captured six spectral bands—blue (475 nm), green (560 nm), red (668 nm), red edge (717 nm), near infrared (840 nm), and thermal infrared (11 µm)—while the RGB camera captured standard red, green, and blue bands in the 400–700 nm range, for a total of nine bands per flight. On-board GNSS and an inertial navigation unit recorded positional data stored in EXIF metadata, as well as light and flux information collected by a solar sensor. The data collection took place in July 2022 in Corvera. The sky was clear. The flight plan, designed using QBase 3D software, provided the real-time monitoring of altitude, distance, battery status, and telemetry. The flight was conducted at an altitude of 190 metres with a ground sampling distance (GSD) of 8.8 cm, covering approximately 30 hectares with 85% frontal and 75% lateral image overlap. Images were automatically acquired, and alignment was in accordance with design parameters. Photogrammetric processing in Pix4D Professional included image orientation, 3D reconstruction, DEM generation, and orthorectification to produce a GeoTIFF mosaic in the EPSG:4326 coordinate system. Prior to orthophoto generation, radiometric calibration was performed. The image values were already normalised to reflectance by dividing each pixel value by 32,768 (equivalent to 100% reflectance). Geometric corrections were performed using ground control points (GCPs) collected by a Differential Global Positioning System (DGPS). Radiometric correction and reflectance transformation were performed using the Grey Scale Correction method.

### 2.4. Sentinel-2 Imagery

The spectral data employed in this study was obtained from the MultiSpectral Instrument (MSI) aboard the Sentinel-2A satellite, part of the Copernicus Sentinel-2 mission. The MSI captures imagery across 13 spectral bands, with spatial resolutions of 10, 20, or 60 m, covering the visible, near-infrared (NIR), and short-wave infrared (SWIR) regions of the electromagnetic spectrum (Table 1). Specifically, bands B2 (Blue, 490 nm), B3 (Green, 560 nm), B4 (Red, 665 nm), and B8 (NIR, 842 nm) offer a 10 m resolution; bands B5 (705 nm), B6 (740 nm), B7 (783 nm), B8A (865 nm), B11 (1610 nm), and B12 (2190 nm) provide a 20 m resolution; while bands B1 (443 nm), B9 (940 nm), and B10 (1375 nm) have a 60 m resolution. The Sentinel-2 images utilised were sourced from the French land data centre, THEIA (https://theia.cnes.fr (accessed on 20 July 2022)). For each acquisition date, ten spectral bands—B2, B3, B4, B5, B6, B7, B8, B8A, B11, and B12—underwent comprehensive pre-processing to correct for atmospheric effects, topographic variations, and sensor-related anomalies. These corrections were performed using the MACCS-ATCOR Joint Algorithm (MAJA) processor, which integrates multi-temporal analysis for enhanced cloud detection and atmospheric correction. The processed data, classified as Level-2A products, represented surface reflectance values that had been corrected for atmospheric disturbances. This study placed particular emphasis on bands B2, B3, B4, and B8 due to their 10-metre spatial resolution and their ability to capture the visible and NIR spectra, which are critical for vegetation and soil analysis. Furthermore, the SWIR region’s sensitivity to moisture content and biomass characteristics was exploited by incorporating bands B11 and B12, which had a 20-metre resolution. To make sure the datasets were all the same, the spectral bands were resampled to a common spatial resolution of 20 metres using the nearest neighbour interpolation method. The study’s objectives in modelling and mapping soil organic carbon and vegetation carbon stocks were underpinned by consistent analysis and interpretation of the multispectral data, facilitated by this harmonisation.

The calculation of environmental indices from the Sentinel-2 and Unmanned Aerial System (UAS) datasets is a widely used method for assessing various environmental aspects, including vegetation health, crop conditions, and water quality. The focus of this study was the investigation of the relationship between these indices and soil organic carbon (SOC) storage. The equations used for these indices are detailed in Table 2. The Sentinel Application Platform (SNAP) software was used to obtain bare soil pixel values at specific sampling sites. This software facilitated the extraction of relevant spectral information necessary for SOC estimation.

### 2.5. Building a Model to Predict Soil Organic Carbon at the Regional Scale

Accurate soil organic carbon (SOC) maps at regional scales require data collection and model calibration at the field level. As shown in the conceptual framework (Figure 2), randomly selected field samples form the basis of the dataset used to construct predictive models that can be scaled up to larger landscapes. The proposed methodology uses machine-learning algorithms to integrate field-measured SOC, drone-based multispectral imagery and satellite observations in order to scale up predictions from the farm level to the regional level [31,32]. The workflow involves collecting SOC measurements using either hand-held sensors or laboratory analysis at representative locations within the farm. This ground-truth data is then spatially linked to high-resolution drone imagery to generate predictive SOC models at the plot or farm level. These models are then extrapolated to a regional scale using coarser satellite imagery (e.g., Sentinel-2). This enables the production of comprehensive soil carbon maps. This process can be updated annually, enabling improved accuracy over time as more ground truth data becomes available [33].

The effectiveness of machine-learning algorithms such as Random Forest (RF), Support Vector Machines (SVM), and Artificial Neural Networks (ANNs) in modelling non-linear relationships between spectral data and soil properties has led to their wide use [34]. When it comes to predicting soil organic carbon (SOC) remotely, vegetation indices (e.g., NDVI, SAVI, and OSAVI) and specific spectral reflection values are often used as reliable proxies for the soil properties beneath the surface [35]. The performance of these models is greatly impacted by the quality of the data preparation pipeline. This pipeline includes data cleansing, which removes errors, missing values, and outliers; data integration, which combines data from multiple sources into a consistent format; feature selection, which identifies and retains the most relevant predictors for accurate and robust model development using techniques such as Pearson correlation analysis and importance ranking. The dataset is usually split into training and test subsets for model validation. A 70/30 split is often used. The training set is used to train the machine-learning algorithm, while the test set is used to evaluate its predictive accuracy. The performance of the model is evaluated using standard performance metrics such as Root Mean Square Error (RMSE), Mean Absolute Error (MAE), and Mean Absolute Percentage Error (MAPE) [36]. The modelling process as a whole is carried out in the Python programming environment, making use of its strong machine-learning libraries (for example, Scikit-learn and TensorFlow). The development of black-box prediction models that can be regularly updated as new data is collected and is supported by this pipeline.

### 2.6. Multiple Linear Regression

Multiple Linear Regression (MLR) has been widely employed to estimate difficult-to-measure soil attributes such as Soil Organic Carbon (*SOC*). The MLR model used in this study is formulated as(1)SOC=α+Xn β+ε
where

-*SOC* represents the soil organic carbon content.-*α* is the intercept.-*X* = (*X*_1_,…,*X_n_*) denotes the vector of remote sensing-derived predictors.-*β* = (*β*_1_,…,*β_n_*) are the model coefficients.-*ε* is the error term.

The assumptions for ε include independence, normality (*ε* ∼ N(0, σ^2^)), zero mean, and constant variance (homoscedasticity). Stepwise regression was applied to select the most relevant predictors using F-statistic *p*-values, and the coefficients were estimated via the least squares method described by Cressie [37].

### 2.7. Random Forest

Random Forest (RF) is a non-parametric ensemble learning algorithm developed by Breiman [38] that is particularly effective for regression and classification tasks. RF uses bootstrap aggregation (bagging) to construct a number of decision trees, combining their outputs to improve predictive performance. A unique advantage of RF is its use of the Gini impurity index to rank variables in order of importance. RF’s robustness to overfitting and its ability to handle non-linear relationships and high-dimensional data make it a preferred choice for SOC modelling [39,40]. Error estimates are obtained using out-of-bag (OOB) samples. This further strengthens the validity of the model [41].

### 2.8. Support Vector Machine

Support Vector Machine Regression (SVR) is a generalisation of the Support Vector Machine (SVM) algorithm for regression tasks. It aims to find a function that deviates from the observed targets by a value no greater than ε, while being as flat as possible [42]. Kernel functions project inputs into higher-dimensional spaces, enabling non-linear relationships to be captured [43]. Figure 3 illustrates the network structure of the SVM algorithm used to predict soil organic carbon (SOC) content.

Of the available kernels, the Radial Basis Function (RBF) kernel was found to provide the highest prediction accuracy. Hyperparameters such as the penalty parameter C and the epsilon ε were optimised using a grid search and tenfold cross-validation.

The objective of the SVR was to find the function *f*(*x*) that had the most deviance from the actual target obtained $y_{i}$ for all training data.

SVR training aims to solve(2)$$\begin{array}{c} Min\frac{1}{2}{\left\|W\right\|}^{2}\\ \mathrm{Subject}\;\mathrm{to}\;\left\{\begin{array}{c} y_{i}-<w_{i}\;x_{i}>-b\leq \varepsilon ,\\ <w_{i}\;x_{i}>+b-y_{i}\leq \varepsilon \end{array}\right.\\ y=<w_{i}\;x_{i}>+b+\varepsilon \end{array}$$

The internal product plus interception is the prediction for this sample and a free parameter that represents a threshold. $x_{i}\;y_{i}<w_{i}\;x_{i}>+b\;\varepsilon$.

### 2.9. Artificial Neural Network (ANN)

Artificial Neural Networks (ANNs), in particular the feed-forward multilayer perceptron (MLP), have been identified as powerful modelling tools for capturing complex non-linear relationships [44]. The MLP network architecture comprises an input layer, one or more hidden layers, and an output layer. Each neuron in the hidden layers processes inputs by applying a weighted combination of values and an activation function, which in this case is the rectified linear unit (ReLU) [45]. In this study, an MLP ANN with input, hidden, and output layers was implemented to predict soil organic carbon (SOC) content based on remote sensing data, as illustrated in Figure 4. The configuration of the ANN model was undertaken in accordance with the approach of Liu and Kuang [46,47]. This entailed the tuning of hyperparameters, including the learning rate (ranging from 0.001 to 0.05), the number of neurons in the hidden layers (2 to 10), and the number of training epochs (10 to 100). This configuration enabled the network to effectively model the non-linear interactions between spectral features and SOC variability [48]. However, the observed underestimation of high SOC values by the ANN model highlights a common limitation in regression-based prediction tasks, especially when the dataset is imbalanced. In this study, high SOC values were underrepresented in comparison to moderate and low values, which is likely to have resulted in biased model learning towards the dominant SOC range. Furthermore, neural networks have been observed to generalise towards the mean when trained on imbalanced data, which may provide a theoretical explanation for their reduced sensitivity in capturing extreme SOC concentrations. While the ANN demonstrated good overall prediction accuracy, this limitation indicates the need for targeted strategies such as data augmentation, weighted loss functions, or ensemble learning methods to enhance model performance in carbon-rich zones. It is imperative to address this issue to ensure reliable SOC mapping in high-carbon soils, which are often critical for land management, restoration, and carbon sequestration efforts.

### 2.10. Performance Evaluation of SOC Prediction Models

The evaluation of soil organic carbon (SOC) prediction models constitutes a fundamental step in the assessment of the effectiveness and reliability of remote sensing and machine-learning approaches for environmental modelling. A variety of statistical metrics can be utilised to quantify the accuracy and robustness of these models. Root mean square error (*RMSE*), mean absolute error (*MAE*), mean absolute percentage error (*MAPE*), and the coefficient of determination (R^2^) are frequently utilised performance indicators. These metrics provide a comprehensive assessment of model performance by comparing predicted SOC values with observed laboratory measurements [49,50,51]. These tools facilitate the quantification of predictive accuracy, the identification of potential sources of bias, and the iterative refinement of the model. Metrics of this nature are of particular value when evaluating complex models based on remote sensing and machine learning. Such models are often prone to overfitting or poor generalisation in heterogeneous agricultural landscapes. The following formulae are employed in order to calculate the key performance metrics in this study:(3)$$RMSE=\sqrt{\frac{\sum_{1}^{n}{\left(X_{p}-X_{a}\right)}^{2}}{n}}$$(4)$$MAE=\frac{1}{n}\sum_{1}^{n}\left|X_{p}-X_{a}\right|$$(5)$$MAPE=\frac{100}{n}\sum_{1}^{n}\left|\frac{\left(X_{p}-X_{a}\right)}{X_{a}}\right|$$
where

$X_{p}$ predicted SOC

$X_{a}$ actual (measured) SOC

*n* = total number of observations.

## 3. Results and Discussion

### 3.1. Summary Statistic

A comprehensive statistical analysis was conducted to characterise the soil organic carbon (SOC) dataset, which comprised 70 soil samples (see Table 3). The primary objective of this analysis was to provide a foundation of knowledge that would subsequently inform and support predictive modelling efforts. The SOC values exhibited considerable variability, with a mean of 40.83 t/ha and a standard deviation of 14.84 t/ha, ranging from 11.88 to 74.62 t/ha. This substantial variation suggests a heterogeneous spatial distribution of SOC within the study area. To further quantify this dispersion, the coefficient of variation (CV) was calculated by dividing the standard deviation by the mean, yielding a CV of approximately 36.3%. This relatively high CV is indicative of significant variability, likely influenced by factors such as land use, soil texture, moisture conditions, and land management practices.

In order to examine the relationships among the input variables and SOC content, a correlation matrix was visualised as a heat map (see Figure 5). This exploratory step was of crucial importance for the identification of multicollinearity among variables and the selection of the most informative predictors for model development. The correlation analysis elucidated the individual relationships between spectral bands/indices and SOC, whilst also revealing interactions among the predictor variables themselves.

The correlation analysis presented in Figure 5 demonstrates a strong positive relationship between soil organic carbon (SOC) and several vegetation indices, particularly NDVI, EVI, SAVI, OSAVI, and DVI, with correlation coefficients ranging from 0.83 to 0.84. The indices, derived from red and near-infrared bands, reflect vegetation vigour and cover. It is thus indicated that areas with denser or healthier vegetation tend to store more organic carbon. Furthermore, a strong correlation was demonstrated by SOC with the NIR band (0.88) and moderate correlations with visible bands (blue: 0.64, green: 0.70, and red: 0.74), thus emphasising its spectral sensitivity to both vegetation and soil characteristics. Conversely, the Brightness Index (BI) exhibited a strong negative correlation with SOC (−0.77), suggesting that soils with higher brightness, likely to be drier or more exposed, tend to have lower carbon content. The high correlation between SOC and measured organic matter (MO: 0.94) further validates the reliability of the dataset. While the heatmap effectively illustrates these relationships, it also reveals high inter-correlation among several vegetation indices, particularly those derived from similar spectral bands. In order to address the issue of potential multicollinearity, Variance Inflation Factor (VIF) analysis was conducted. Variables with VIF values exceeding a threshold of 10 were considered to be highly collinear. On the basis of their predictive significance, these were either removed or merged. This step was essential to reduce redundancy in the input data, improve model performance, and enhance the interpretability and robustness of SOC predictions across heterogeneous landscapes.

### 3.2. Multiple Linear Regression

Figure 6 shows the results of a multiple linear regression model that was used to predict soil organic carbon (SOC) based on ten input features. These features included spectral bands (blue, green, red and near-infrared [NIR]) and vegetation indices: the Normalised Difference Vegetation Index (NDVI), Enhanced Vegetation Index (EVI), Optimised Soil-Adjusted Vegetation Index (OSAVI), Soil-Adjusted Vegetation Index (SAVI), Difference Vegetation Index (DVI), and Brightness Index (BI). The scatter plot compares the actual and predicted SOC values, with the blue dashed line representing the ideal 1:1 fit (perfect prediction). However, the model’s predictive performance was poor, as indicated by a negative R^2^ value of −1.22 and a relatively high RMSE of 20.64. A negative R^2^ value means that the model performed worse than predicting the mean of the observed values. The substantial scatter around the ideal line suggests that the linear model failed to capture the complex, potentially non-linear relationships between the input features and SOC. This highlights the limitations of using simple linear regression for SOC estimations in heterogeneous environments.

These results suggest that linear regression may be unsuitable for modelling SOC in this context, likely due to non-linear relationships between SOC and remote sensing inputs.

### 3.3. Support Vector Regression

Table 4 compares the performance of SVR models for predicting soil organic carbon (SOC) using three types of data normalisation (default, minimum–maximum, and robust) and two data sources: standard, min–max, and robust: UAV (drone images) and Sentinel-2 satellite images.

The evaluation is based on the coefficient of determination (R^2^) and mean square error (MSE). The results show that models trained with UAV data performed significantly better than models trained with Sentinel-2 data. UAV-based models achieved R^2^ values up to 0.47 and lower MSE values (around 26.48 to 27.55), with the Robust_Scaler providing the best accuracy on both training and test sets. In contrast, models based on Sentinel-2 data showed poor predictive performance, with negative R^2^ values (as low as −0.40) and very high MSE (over 266). This indicates their limited ability to explain SOC variance, probably due to the coarser spatial resolution of Sentinel-2 (10–20 m) compared to UAV imagery. Among the scaling methods, the Robust_Scaler appeared to be the most effective, particularly for UAV data, due to its resilience to outliers. UAV imagery in combination with robust normalisation is the most reliable approach for SOC estimation using SVR, while Sentinel-2 data is not suitable for accurate SOC prediction under the conditions tested.

Figure 7 shows a visual comparison of actual and predicted soil organic carbon (SOC) values. Three different data normalisation techniques were used: Standard_Scaler, MinMax_Scaler and Robust_Scaler.

The actual SOC values are shown as solid lines for each scaler. The predicted values are shown as dashed lines in corresponding colours. The graph clearly shows that for all three scaling methods, the predicted values remained relatively flat and did not follow the strong variability observed in the actual SOC measurements. This discrepancy suggests a significant underfitting problem. The SVR models were unable to capture the true complexity of the SOC distribution. All scalers performed poorly. However, the Robust_Scaler (purple) performed slightly better than the others, particularly in capturing subtle trends. However, none of the models were able to reproduce the sharp peaks and troughs in the SOC data. This highlights that additional feature engineering, model tuning, or higher resolution input data may be required to improve prediction accuracy.

Derived from the test dataset, the outcomes reveal a minor overfitting tendency within the Random Forest regression model, as illustrated in Figure 8.

### 3.4. Random Forest

The coefficient of determination (R^2^), approximately 0.85, signifies that the model accounted for roughly 85% of the variance within the test data. Although this still implies a substantial correlation between the predicted and actual values, the model’s performance exhibited a slight reduction in comparison to the training set.

Figure 9 illustrates the comparison between actual and predicted soil organic carbon (SOC) values derived from Sentinel-2 satellite data during the training phase.

The blue line represents the observed SOC values. The orange line corresponds to the model predictions. The graph shows a noticeable gap between actual and predicted values, particularly at several peaks (e.g., indices 0, 5, and 6), where the model significantly underestimated the SOC. This underfitting highlights the limited ability of the model to fully capture the variability of SOC observed in the field, likely due to limitations related to the quality of the input variables, the complexity of the model, or the spatial resolution of the Sentinel-2 data. While the overall trend of SOC variation was moderately followed, the prediction failed to accurately capture higher SOC values. This is reflected in the model’s performance metrics, with an R^2^ of 0.65 and a root mean square error (RMSE) of 30.86, indicating a moderate fit with significant error. These results suggest that, although Sentinel-2 satellite data can be useful for broad, regional-scale assessments, its effectiveness at finer spatial scales is limited. Improving the model may require better pre-processing, the integration of additional variables (e.g., soil texture and elevation), or more advanced modelling approaches, such as ensemble methods or deep learning. By contrast, UAV-based remote sensing offers superior spatial resolution and predictive accuracy, making it better suited to detailed parcel-level soil organic carbon (SOC) mapping. While both datasets offer valuable insights, UAV-based remote sensing provides superior spatial detail and accuracy for soil organic carbon (SOC) modelling. Though the Sentinel-2 model is useful for regional-scale assessments, it may require additional pre-processing, the integration of ancillary data, or model enhancements (e.g., advanced feature engineering or ensemble techniques) to improve its performance at finer spatial scales.

### 3.5. Artificial Neural Network

The Artificial Neural Network (ANN) model developed for the prediction of soil organic carbon (SOC) from multispectral features (blue, green, red, NIR, NDVI, EVI, OSAVI, SAVI, DVI and BI) showed moderate to good prediction performance. The model illustrated in Figure 10 achieved an R^2^ value of about 0.68, which means that it explains about 68% of the variability in SOC values. The RMSE and MAE are quite low, indicating good but not perfect accuracy. The comparison graph between predicted and actual SOC values shows that the ANN captured the general trends well, although it tended to slightly underestimate high SOC values and overestimate low SOC values, suggesting some smoothing effects, probably due to data distribution and model limitations. Overall, the ANN demonstrated robust capabilities for predicting SOC across agricultural fields. However, further improvements could be achieved by expanding the dataset, especially for extreme values, and by including additional environmental variables or exploring more complex model architectures.

The UAV-based model shows significantly better performance in soil organic carbon (SOC) prediction than Sentinel-2 data at all evaluation stages (training, selection and testing).

UAV imagery, due to its finer spatial resolution, captured soil variability more effectively, resulting in lower SSE, MSE, and RMSE values and positive NSE values, indicating a good model fit. Conversely, the Sentinel-2 model had much higher errors (SSE and RMSE) and negative NSE values. This reflects poor model performance, where predictions were less accurate than simply using the mean SOC. These results highlight the advantage of high-resolution drone imagery for the accurate estimation of the SOC compared to satellite-based approaches such as Sentinel-2.

The graph shown in Figure 11 compares laboratory measurements of soil organic carbon (SOC) with predictions made using four machine-learning models: soil organic carbon (SOC), soil organic carbon (SOC), and soil organic carbon (SOC): Random Forest, Multiple Linear Regression (MLR), Support Vector Machine (SVM), and an Artificial Neural Network (ANN). The black dashed line represents perfect agreement (1:1 line). Deviations from this line indicate prediction errors. ANN and SVM show low variability but tend to consistently underestimate SOC, while Random Forest covers a wider range with some overestimation at higher SOC levels. The MLR shows poor performance. Its predictions are both unstable and imprecise. Overall, there is little agreement between the models and the measured SOC, highlighting the presence of prediction bias and error, particularly at extreme values. Further improvements could be achieved through better feature selection, normalisation and advanced modelling strategies. This graph compares laboratory-measured soil organic carbon (SOC) values with those predicted by four machine-learning models: Random Forest, Multiple Linear Regression (MLR), Support Vector Machine (SVM) and an Artificial Neural Network (ANN). The black dashed line represents perfect agreement (the 1:1 line), and deviations from this line indicate prediction errors. While ANN and SVM showed low variability, they consistently underestimated SOC; Random Forest, on the other hand, covered a wider range, with some overestimation at higher SOC levels. MLR showed poor performance, providing unstable and inaccurate predictions. Overall, none of the models showed perfect agreement with the measured SOC values, highlighting the presence of prediction bias and error, particularly at extreme values. Further improvements could be achieved through better feature selection, normalisation and advanced modelling strategies.

The findings of this study highlight significant variations in the efficacy of UAV and Sentinel-2 remote sensing data, in conjunction with various machine-learning models, for predicting soil organic carbon (SOC) across agricultural landscapes. In accordance with the findings of preceding research, UAV-based data exhibited superior performance in comparison to Sentinel-2 imagery across the full range of modelling approaches, primarily as a consequence of its superior spatial resolution and capacity to discern small-scale heterogeneity in vegetation and soil conditions [52,53]. This finding is consistent with the results reported by Beltrán-Marcos et al. [19], who demonstrated that UAV imagery significantly enhanced SOC estimates in post-fire forest landscapes when compared to Sentinel-2.

In the course of the experimental process, it was found that Support Vector Regression (SVR) demonstrated a substandard performance, particularly in relation to Sentinel-2 data, as evidenced by negative R^2^ values and elevated mean square error (MSE) metrics. This under-performance is indicative of the known limitations of SVR when working with coarse-resolution data and limited feature diversity and aligns with previous studies showing that SVR often requires extensive parameter tuning and feature engineering to achieve optimal results [54].

Conversely, the Random Forest (RF) algorithm delivered the most accurate results using UAV data, achieving an R^2^ of 0.85. This finding serves to substantiate the model’s efficacy in managing intricate, non-linear interactions between remote sensing variables and SOC, particularly in the context of fine spatial resolutions. However, the performance of RF declined with Sentinel-2 inputs, particularly for high SOC values. This was likely due to the effects of spatial averaging and spectral mixing, which are inherent in medium-resolution data. These findings are consistent with the conclusions of Yang et al. [55] and Abbaszad et al. [56], who reported superior RF performance using high-resolution imagery for SOC mapping.

The Artificial Neural Network (ANN) model produced reliable predictions of the SOC with an R^2^ value of approximately 0.68. This demonstrates its ability to capture general SOC trends while smoothing out extreme values. While the ANN demonstrated a marked improvement in performance with UAV data, it also exhibited moderate success with Sentinel-2 data, thereby reflecting its capacity for adaptation to diverse data sources. However, the observed smoothing effect suggests that further improvements could enhance the model’s accuracy. Such improvements could include expanding the dataset to cover more extreme SOC values and testing more advanced architectures [56]. The performance of Artificial Neural Networks (ANNs) was moderate to strong (R^2^ ≈ 0.68), demonstrating adaptability to both UAV and Sentinel-2 data. However, the ANN tended to underestimate high SOC values, a phenomenon that has also been observed in other SOC modelling efforts [57]. This finding suggests the necessity for further enhancement, such as the incorporation of deeper network architectures, more diverse training data, or ensemble methods to improve sensitivity to extreme values.

From a methodological perspective, Sentinel-2′s moderate spatial resolution (10–20 m) presents significant limitations, especially in heterogeneous agricultural fields. The reliability of SOC estimates is reduced by spectral mixing from vegetation residues, moisture variability, and soil surface conditions, as evidenced by the findings of Tan et al. [58] and Zhang et al. [39]. Furthermore, the Sentinel-2 satellite does not possess short-wave infrared (SWIR) bands, which have been shown to exhibit heightened sensitivity to soil organic carbon (SOC). This spectral limitation reduces the capacity to isolate SOC-specific reflectance features.

Another significant constraint pertains to the reliance of machine-learning models on substantial, balanced, and representative training datasets. Although the field sampling in this study was detailed, it covered a limited number of high SOC zones, which resulted in the underrepresentation of extremes. Furthermore, the absence of auxiliary soil variables (e.g., pH, texture, and moisture) was likely to have constrained model performance, especially for SVR and ANN models, which are known to be sensitive to feature space completeness.

The comparative analysis in Table 5 confirms that UAV-based models outperformed Sentinel-2 across all evaluation metrics (e.g., SSE, RMSE, and NSE), with UAV-derived RF models achieving the lowest error rates and positive NSE values. These results align with recent studies that emphasise the superior spatial precision of UAV imagery for SOC estimation at local scales [53,59]. However, Sentinel-2 remains valuable for regional-scale assessments. With further enhancements, such as data fusion, inclusion of radar (e.g., Sentinel-1), or SWIR bands, and ensemble modelling (e.g., XGBoost), it could be made more competitive [60,61].

In summary, the present study is consistent with and builds upon extant literature by confirming the superior performance of UAV data for fine-scale SOC modelling and highlighting the limitations of Sentinel-2 for field-scale analysis. The integration of UAV imagery with robust machine-learning algorithms, such as Random Forest and ANN, presents a promising pathway for high-resolution SOC mapping. It is recommended that future research endeavours concentrate on incorporating additional environmental variables, expanding the temporal and spatial extent of training datasets, and exploring advanced modelling frameworks, such as ensemble and geographically weighted learning, with a view to enhancing prediction accuracy across scales and landscapes.

## 4. Conclusions

This study corroborated the hypothesis that UAV-based multispectral imagery, due to its superior spatial resolution, significantly outperforms Sentinel-2 satellite data in predicting soil organic carbon (SOC) when integrated with machine-learning models. Among the algorithms that were tested, Random Forest (RF) achieved the highest prediction accuracy, especially with UAV data, while Artificial Neural Networks (ANNs) also demonstrated moderate success in capturing general SOC trends. Sentinel-2′s performance, whilst beneficial at regional scales, was constrained by spectral mixing and its limited sensitivity to SOC-specific reflectance features. Notwithstanding the encouraging results obtained, it was possible to identify several limitations. Firstly, the moderate spatial resolution of Sentinel-2 (10–20 m) limited its capacity to capture fine-scale variability in heterogeneous agricultural landscapes. Secondly, the study relied exclusively on spectral features, neglecting the incorporation of crucial soil covariates such as pH, texture, and moisture, which are recognised as influencing SOC distribution. Thirdly, the limited number of samples representing high SOC values may have affected the model’s ability to capture extreme conditions, particularly in the ANN, which showed a tendency to underestimate upper-range SOC values. Furthermore, the utilisation of a straightforward train–test split, devoid of k-fold cross-validation, may have engendered concerns regarding overfitting or diminished generalizability. In order to address the current limitations of the research, future studies should be conducted with larger sample sizes and greater geographic scope. Additionally, the inclusion of additional soil and environmental variables, as well as data fusion techniques such as UAV and Sentinel integration, is recommended. Furthermore, the exploration of advanced modelling techniques, including deep learning, is advised. The implementation of these measures is expected to enhance the accuracy, scalability, and applicability of the SOC model across diverse geographical areas, thereby facilitating optimised land and climate management.

## Figures and Tables

**Figure 1 sensors-25-05281-f001:**
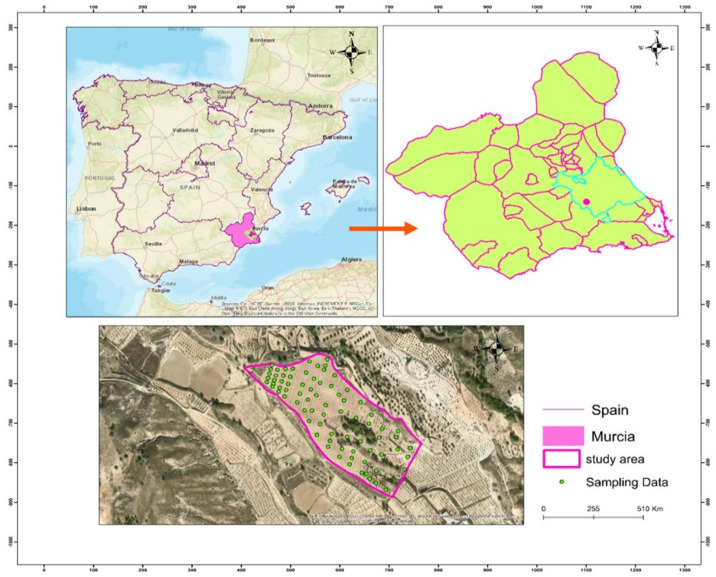
Maps of the study area in Murcia watershed and its sampling points.

**Figure 2 sensors-25-05281-f002:**
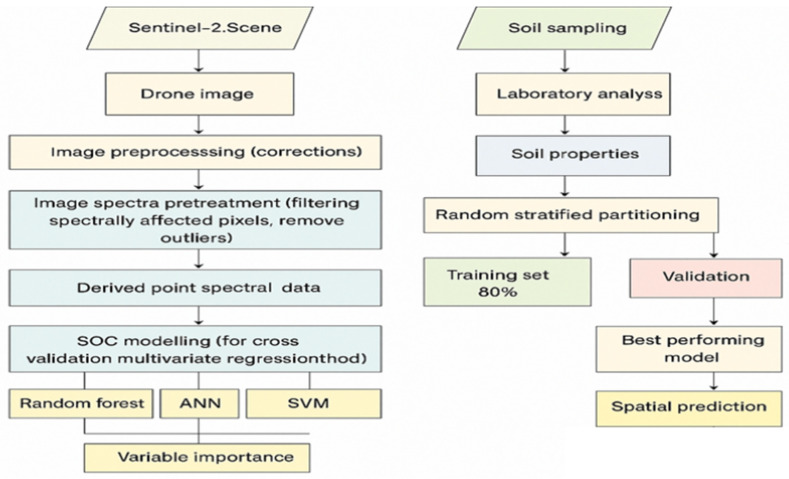
Flow chart showing the soil carbon mapping from field to regional scale using artificial intelligence. Modelling MLR, SVM, Random Forest, and ANN.

**Figure 3 sensors-25-05281-f003:**
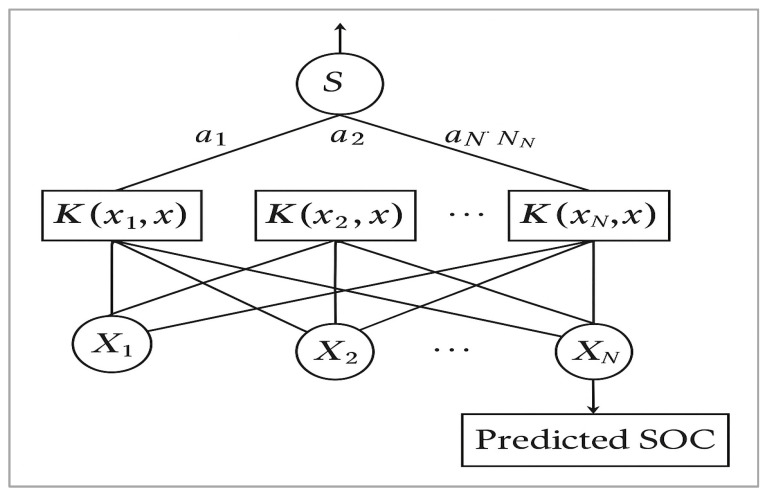
Structure of SVM.

**Figure 4 sensors-25-05281-f004:**
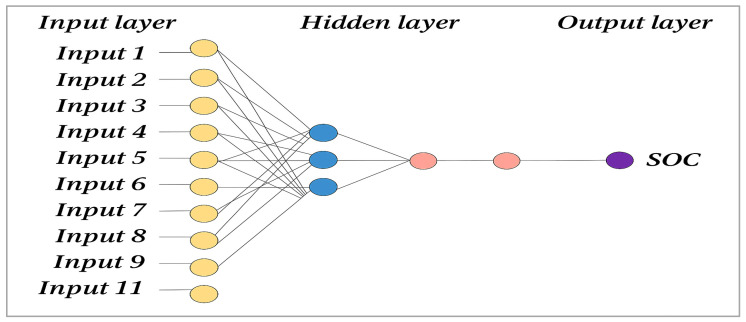
Structure of ANN model.

**Figure 5 sensors-25-05281-f005:**
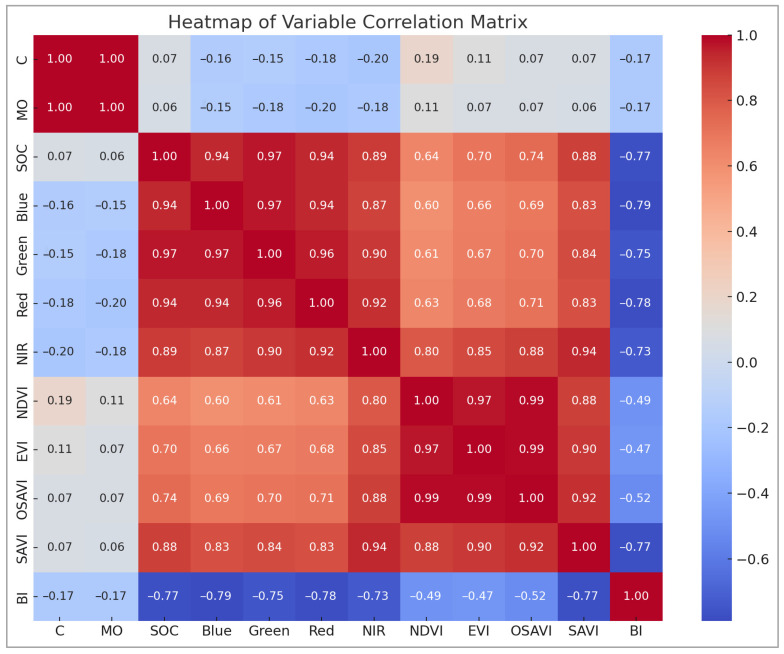
Heat map correlation between different variables.

**Figure 6 sensors-25-05281-f006:**
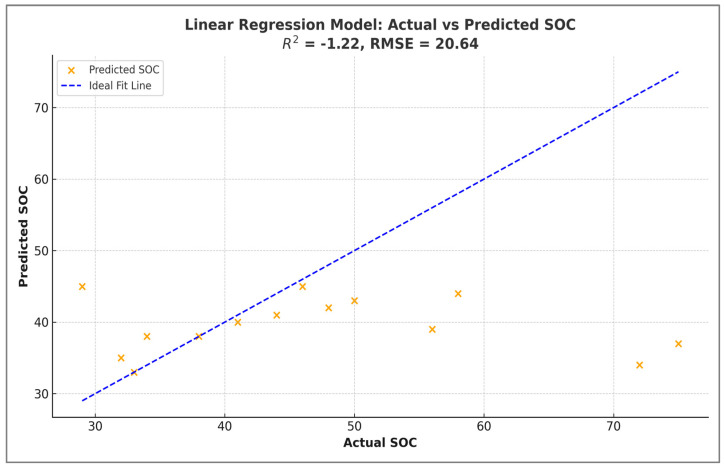
Performance of linear regression model for SOC estimation from Sentinel-2 imagery: Actual vs. predicted values.

**Figure 7 sensors-25-05281-f007:**
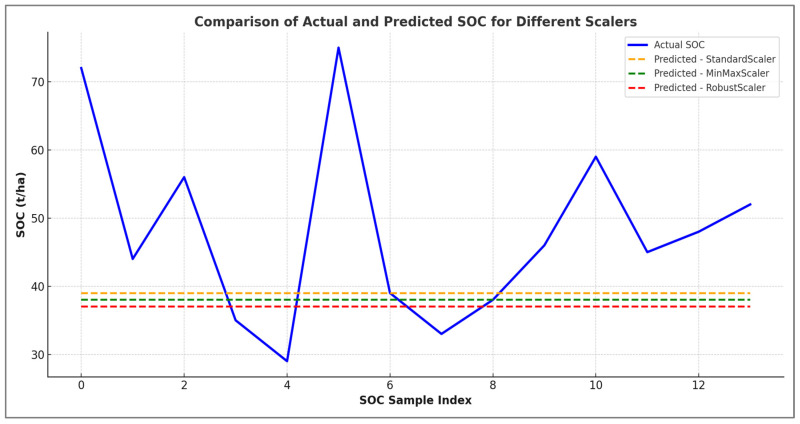
Comparison of actual and predicted SOC for different scalers.

**Figure 8 sensors-25-05281-f008:**
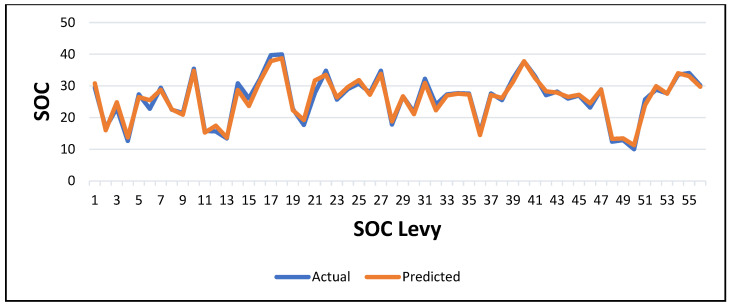
Variation in predicted SOC values from original data UAV (training).

**Figure 9 sensors-25-05281-f009:**
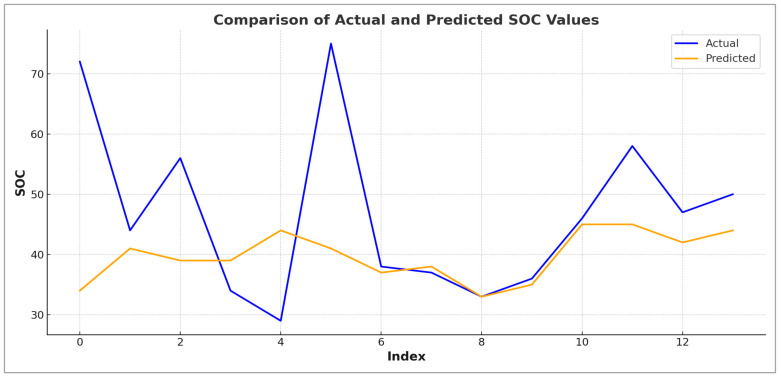
Variation in predicted SOC values from original data Sentinel-2 (training).

**Figure 10 sensors-25-05281-f010:**
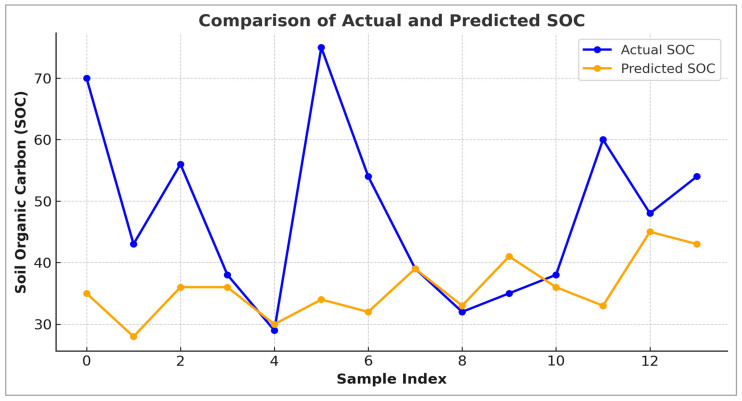
Performance of the ANN model using Sentinel-2 data: comparison between actual and predicted SOC values (test dataset).

**Figure 11 sensors-25-05281-f011:**
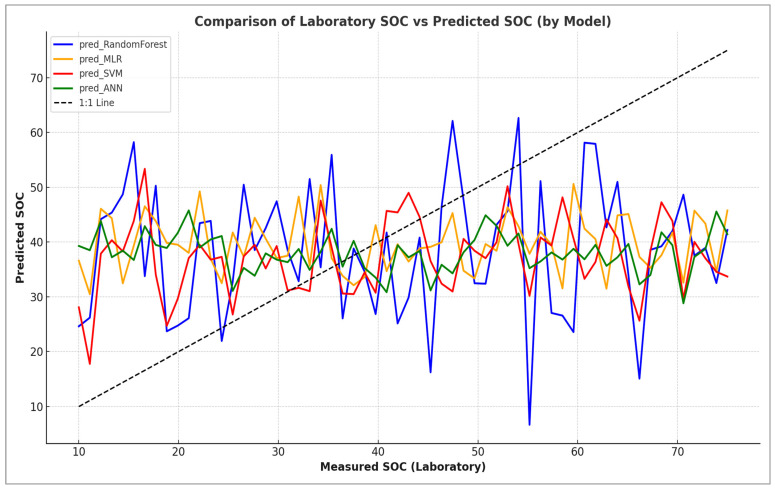
Laboratory vs. predicted soil organic carbon (SOC) across machine-learning models.

**Table 1 sensors-25-05281-t001:** Bands used from the Sentinel-2 multispectral sensor. Determination of environmental indices of drone image and satellite.

Band	Resolution	Central Wavelength	Description
B1	60 m	443 nm	Ultra Blue (Coastal and Aerosol)
B2	10 m	490 nm	Blue
B3	10 m	560 nm	Green
B4	10 m	665 nm	Red
B5	20 m	705 nm	Visible and Near Infrared (VNIR)
B6	20 m	740 nm	Visible and Near Infrared (VNIR)
B7	20 m	783 nm	Visible and Near Infrared (VNIR)
B8	10 m	842 nm	Visible and Near Infrared (VNIR)
B8a	20 m	865 nm	Visible and Near Infrared (VNIR)
B9	60 m	940 nm	Short Wave Infrared (SWIR)
B10	60 m	1375 nm	Short Wave Infrared (SWIR)
B11	20 m	1610 nm	Short Wave Infrared (SWIR)
B12	20 m	2190 nm	Short Wave Infrared (SWIR)

**Table 2 sensors-25-05281-t002:** Determination of environmental indices from drone imagery and Sentinel-2 Data (UAV data acquired on July 20).

Index	Definition Based on Drone Imagery	Definition Based on Sentinel-2	Reference
NDVI	$\frac{\;NIR-R}{NIR+R}$	$\frac{\;B8-B4}{B8+B4}$	[25]
EVI	$\frac{\;2.5\;(NIR-Red)}{\left(NIR+6\;*\;Red-7.5\;*\;Blue+1\right)}$	$\frac{\;2.5\;(B8-B4)}{\left(B8+6\;*\;B4-7.5\;*\;B2+1\right)}$	[26]
SAVI	$\frac{\;\left(NIR-Red\right)}{\left(NIR+Red+L))\;*\;(1+L\right)}$	$\frac{\;\left(B8-B4\right)}{\left(B8+B4+0.5))\;*\;(1+0.5\right)}$	[27]
*DVI*	*NIR–Red*	*B8–B4*	[28]
OSAVI	$\frac{\;\left(NIR-Red\right)}{\;(NIR+Red+0.16)}$	$\frac{\;\left(B8-B4\right)}{\;(B8+B4+0.16)}$	[29]
BI	$\frac{\sqrt{Red^{2}+Green^{2}}}{2}$	$\frac{\sqrt{B4^{2}+B3^{2}}}{2}$	[30]

**Table 3 sensors-25-05281-t003:** Descriptive statistics of all parameters.

	SOC	Blue	Green	Red	Nir	NDVI	EVI	OSAVI	SAVI	DVI	BI
count	70	70	70	70	70	70	70	70	70	70	70
mean	40.83	0.26	0.32	0.40	0.50	0.05	0.11	0.08	0.232	0.09	0.47
std	14.84	0.015	0.020	0.02	0.018	0.01	0.01	0.01	0.02	0.01	0.03
min	11.88	0.2216	0.2687	0.33	0.4512	0.006	0.08	0.06	0.18	0.0725	0.38
0.25	29.001	0.25	0.3182	0.391	0.4898	0.04	0.1	0.07	0.21	0.08	0.45
0.5	39.09	0.2613	0.3292	0.41	0.50	0.052	0.11	0.083	0.23	0.09	0.47
0.75	48.69	0.2707	0.3451	0.43	0.5117	0.05	0.12	0.09	0.24	0.09	0.49
max	74.62	0.3178	0.4082	0.4841	0.558	0.09	0.18	0.13	0.31	0.134	0.56

**Table 4 sensors-25-05281-t004:** Compares the performance of SVR (Support Vector Regression) models on UAV and Sentinel-2.

Scaler	Accuracy on Tested Data				Accuracy on Training Data			
	UAV		Sentinel-2		UAV		Sentinel-2	
	R^2^	MSE	R^2^	MSE	R^2^	MSE	R^2^	MSE
Standard	0.28	55.27	−0.39	268.18	0.46	26.72	0.020	207.007
Min–Max	0.07	58.61	−0.40	269.26	0.45	27.55	0.012	208.77
Robust	0.20	50.31	−0.39	266.30	0.47	26.48	0.021	206.88

**Table 5 sensors-25-05281-t005:** Comparison of UAV and Sentinel-2 model performance for SOC.

	Training		Selection		Testing	
	UAV	Sentinel-2	UAV	Sentinel-2	UAV	Sentinel-2
SSE	203.707	5518.195	117.128	4321.564	116.562	4400.120
MSE	0.038	14.8068	0.066	11.836	0.066	12.230
RMSE	0.196	19.8534	0.258	19.161	0.257	19.244
NSE	0.343	−1.0576	0.337	−0.942	0.32	−0.910

## Data Availability

Data will be made available on request.

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
