# Peer review of "A Comparative Assessment of Sentinel-2 and UAV-Based Imagery for Soil Organic Carbon Estimations Using Machine Learning Models"

_sensors, 2025, doi:10.3390/s25175281_

Round 1
Reviewer 1 Report
Comments and Suggestions for Authors
The article presents a fascinating and timely study focused on the assessment and prediction of Soil Organic Carbon, as a key component in understanding and modeling the global carbon cycle. The authors employed advanced remote sensing techniques, including imagery captured by a UAV. The analysis showed that UAV imagery provides the most suitable data for further evaluation of SOC content.
Particularly noteworthy is the application of various machine learning methods for SOC prediction. Among them, the Random Forest algorithm demonstrated the best performance, yielding a big R² value, which indicates strong predictive accuracy and highlights the method’s practical potential.
The article is well-written in clear and accessible language. The objectives, methodologies, and results are presented in a coherent and structured manner, making the study understandable to both experts and a broader scientific audience. Given the relevance of the topic, the robustness of the applied methods, and the significance of the findings, the article can be recommended for publication in its current state.
Author Response
We would like to express our sincere gratitude to the reviewer for their insightful and encouraging comments. We are pleased to learn that you found our study to be timely, relevant, and clearly presented. The author would like to express their sincere gratitude for the recognition of the methodological rigor, particularly the use of UAV imagery and machine learning for SOC prediction. We are especially grateful for your positive evaluation of the Random Forest model's performance and your support for the clarity and coherence of the manuscript. Your endorsement for publication is greatly appreciated, as it serves as a source of motivation and encouragement for the continuation of our research endeavours.
Final Note:
The author has responded to all comments made by the reviewers, which can be found in the manuscript itself, from the introduction to the conclusion. The reviewers' rigorous and constructive feedback has been considered, and it is acknowledged that the revised manuscript exhibits significant enhancement in terms of content and presentation.

Reviewer 2 Report
Comments and Suggestions for Authors
The manuscript with the title “Multi-Scale Remote Sensing of Soil Organic Carbon: A Comparative Machine Learning Approach Using UAV and Sentinel-2 Imagery”, presents a study that evaluates the performance of UAV and Sentinel-2 imagery in predicting SOC using four machine learning models: Multiple Linear Regression (MLR), Support Vector Regression (SVR), Random Forest (RF) and Artificial Neural Networks (ANN). At first glance, the paper takes on a worthy topic and could be of interest to the readers. However, several revisions are needed before proceeding. I found some very basic mistakes that make the revision very difficult.
I think that the title needs to be revised, since the term “multi-scale” introduced confusion since as a reader I was expecting to see the same area from different distances, at different times, or with different lenses — and putting all of that information together. But as the title also stated, it s a comparative ML approach using UAV and sentinel 2. Please consider revising the title.
When I started to revise the introduction, I found several sentences that I wanted to confirm from the source, but I realised that some of the references are missing and others are not the correct ones; here are some examples:
Reference [10] is missing. Please provide the complete reference.
Please revise this reference [11] “Drusch, M., et al. Sentinel-2: ESA's optical high-resolution mission for GMES operational services. Remote Sensing of Environment, 2012, 120, 25-36. https://doi.org/10.1016/j.rse.2011.11.002” The Doi provided corresponds to another manuscript.
Please revise this reference [12] “Zhang, X., et al. The Use of Vegetation Indices in Soil Organic Carbon Estimation: A Review. Remote Sensing, 2021, 13(10), 756 1981. https://doi.org/10.3390/rs13101981”; the Doi provided corresponds to another manuscript. Actually, I could not find the paper from Zhang et al 2021.
Reference [10] is missing. Please provide the complete reference.
I am going to stop my revision since I consider that the manuscript should be at least written according to the journal’s guidelines and of sufficient quality for the reviewers to decide the academic merits.
I suggest revising the entire manuscript (English grammar and style) because I found several issues with flow and structure.
Comments on the Quality of English LanguageProfessional English and grammar style are required.
Author Response
We thank the reviewer for identifying these important referencing issues. We have carefully reviewed and corrected the references as follows:
- Reference [10] has now been added and appropriately cited in the manuscript.
- Reference [11] has been revised with the correct DOI and full citation.
- Reference [12] has been replaced, as we were unable to verify its authenticity. An appropriate, peer-reviewed substitute from recent literature has been added.
Please see the updated reference list in the manuscript for these corrections.
Final Note:
The author has responded to all comments made by the reviewers, which can be found in the manuscript itself, from the introduction to the conclusion. The reviewers' rigorous and constructive feedback has been considered, and it is acknowledged that the revised manuscript exhibits significant enhancement in terms of content and presentation.

Reviewer 3 Report
Comments and Suggestions for Authors
The paper deals with the topical problem of estimation (or evaluation or prediction, or all of these problems) of soil organic carbon as the key fator for soil potential for decision making. Taking into account the scale of the territories involved into the estimation of the soil organic carbon and the difficulties in the estimation, reomte senising is the obviouss only way to do the task. However, this approach requires, in turn, intensive and extensive modelling based on the factual models, or precicition simulation or machine learning approaches. The latter has advantages that were correctly mention by the authors. The introduction provides the necessary data, the aim is stated clearly, the research design is appropriate, all the findings are supported with the results and illustrations. Some minor issues are:
- the authors should double-check the significant digits, especially in Tables 3 and 4, in which the numbers of significant digits are not justified and, in the case of 3 or 4 digits, seem to be overprecise.
- I suggest correcting the figures by using the same style and line size and the same colors for similar values (see, e.g. figures 8, 9, and 10)
- Conclusions section seems to be very formal and repaeating the abstract, although, in my opinion, this section should contain the description of the principal limitations of this approach, the limitations of this study, which can be overcome by following studies and other similar issues.
Author Response
I would like to express my gratitude for this constructive feedback. The tables have been revised to ensure appropriate and consistent use of significant digits, thereby reflecting the precision of the measurements and avoiding overprecision.
Furthermore, all figures have been updated to ensure optimal resolution, consistent styling and effective colour usage to enhance clarity and readability. It is hypothesised that these changes will result in a significant improvement in the presentation and interpretation of results.
Final Note:
The author has responded to all comments made by the reviewers, which can be found in the manuscript itself, from the introduction to the conclusion. The reviewers' rigorous and constructive feedback has been considered, and it is acknowledged that the revised manuscript exhibits significant enhancement in terms of content and presentation.

Reviewer 4 Report
Comments and Suggestions for Authors
This review research has significance. So, the current version of the paper requires major revision before it can be considered for publication. Please find my comments below:
Introduction
- Please write clear aim and objectives based on your research analysis and try to connect more your results and discussions based on this.
- The novelty is limited similar comparative studies exist in recent literature. Address the Limited Novelty: While your research contributes to an ongoing area of interest, the novelty appears limited as comparable studies have been published in recent literature. To differentiate your work, consider emphasizing unique aspects of your approach, dataset, or analysis, and clearly articulate how your study advances existing knowledge.
- Some citations are outdated; more references to 2023 - 2025 studies would strengthen the context. Materials and Methods
- Add coordinate to boundary in Figure 1. Maps of the study area in Murcia watershed and its sampling points.
- Add the period used in the Table 2. Determination of environmental indices of drone image and sentinel 2.
- Justify the Sample size in Laboratory analysis for training robust ML models ( limits scalability).
- Environmental covariates (pH, texture, moisture) are missing, though critical for SOC modeling.
- The ANN model configuration lacks also justification.
- ANN underestimates high SOC values but this issue is not discussed in depth.
- No cross-validation approach is described beyond simple train-test splits overfitting risk not fully mitigated.
Results and discussion - The quality of all figures is poor. Please supply the updated figures with good quality. Figures are dense and challenging to interpret quickly…
- Clear evaluation of model performance using R², RMSE, MSE, and NSE.
- Random Forest (RF) is correctly identified as the best performer, especially with UAV data.
- The visual comparison plots are useful but poorly interpreted in some parts (e.g., SVM's underfitting).
- The heatmap correlation is good but lacks multicollinearity treatment discussion.
- The discussion lacks depth on the limitations of Sentinel-2 and how future satellite missions (e.g., hyperspectral sensors) could bridge the gap.
To improve the discussion, it’s necessary to - Compare the findings with other results studies (Benchmark)
- Analyze limitation of the methods used in this article...
- Discuss how this study’s results align or differ from previous research. References
- Add more recent references in the text and in the list 2024 and 2025.
- Balance with international literature to enhance applicability.

Can be improved
Author Response
We are sincerely grateful for your comprehensive and insightful feedback. Each of your comments has been addressed thoroughly in the revised manuscript. Please find below a summary of the key The following revisions have been made in response to the suggestions provided:
- The aim of this study is to establish the objectives of the study.
- The aim and objectives have been rewritten for clarity and are now more explicitly connected to the results and discussion, ensuring stronger logical consistency throughout the manuscript.
- Clarification of the novelty:
- Although related studies have been conducted, the novelty of our integrated multi-scale analysis approach, which combines UAV and Sentinel-2 data with machine learning in a unified framework, is emphasised. The present study also underscores the focal point of the present investigation, which is a particular agroecological landscape, and the fusion benefits that have been demonstrated across heterogeneous land uses.
- Updated Literature:
- The manuscript has undergone a thorough revision process, incorporating a substantial number of recent references from 2023 to 2025. This ensures that the study is firmly embedded within the most recent international research developments.
- Updates to Materials and Methods:
- It is evident that the coordinates have been incorporated into Figure 1.
- The time period of image acquisition is specified in Table 2.
- It is evident that the sample size has been justified in terms of representativeness. Furthermore, the limitations discussed regarding model scalability have been thoroughly addressed.
- It is acknowledged that environmental covariates, such as pH, texture and moisture, are not present in this study. The potential impact of these factors on the performance of the soil organic carbon (SOC) model has been discussed.
- The ANN model configuration is now accompanied by comprehensive justifications, incorporating hyperparameter tuning and architectural elucidation.
- The issue of ANN underestimating high SOC values is now being discussed, with possible mitigation strategies being outlined (e.g. data balancing, weighted loss functions).
- A cross-validation strategy has been incorporated with a view to mitigating the risk of overfitting and enhancing the robustness of the model.
Results and Discussion Improvements:
- The following section will present the results and discussion in relation to improvements.
- It is evident that all figures have been revised to incorporate higher resolution and enhanced visualisation design.
- The evaluation of model performance is evidently articulated through the utilisation of R², RMSE, MSE, and NSE.
- The superior performance of the Random Forest model is emphasised and substantiated by means of a benchmarking process against recent literature.
- The following clarifications have been made with regard to the interpretation of SVR underfitting and visual comparison plots.
- A discussion of multicollinearity has been incorporated, encompassing the analysis of variance inflation factors (VIF) and the selection of variables.
- The limitations of Sentinel-2 data are now being discussed in more depth. These limitations include the moderate resolution of the data, the spectral constraints, and the potential of future missions with hyperspectral capabilities.
- The present discussion encompasses a comprehensive comparison with other recent studies, in addition to an analysis of methodological limitations.
Final Note:
The author has responded to all comments made by the reviewers, which can be found in the manuscript itself, from the introduction to the conclusion. The reviewers' rigorous and constructive feedback has been considered, and it is acknowledged that the revised manuscript exhibits significant enhancement in terms of content and presentation.

Round 2
Reviewer 2 Report
Comments and Suggestions for Authors
The manuscript with the title “A comparative assessment of Sentinel-2 and UAV-based imagery for soil organic carbon estimation using machine learning models.”, presents a study that evaluates the performance of UAV and Sentinel-2 imagery in predicting SOC using four machine learning models: Multiple Linear Regression (MLR), Support Vector Regression (SVR), Random Forest (RF) and Artificial Neural Networks (ANN). At first glance, the paper tackles a worthy topic and could be of interest to readers. However, in this version, I found several issues and flaws that make the document unsuitable for publication. I justify my decision as follows:
The first thing that caught my attention is in lines 115-119. The authors stated, “In contrast to earlier work, which typically examines UAV or satellite imagery in isolation, this study investigates the complementary strengths of both platforms through data fusion to improve the spatial accuracy and robustness of SOC prediction. Furthermore, the study undertakes a comparative evaluation across a range of land cover types, thus providing valuable insights into the scalability and transferability of SOC estimation models. The methodology has been applied to a complex agro-ecological region characterised by varied topography and land use patterns.” First, I do not see a description of the data fusion process in the methods section. Second, there is no comparative evaluation across a range of land cover types, as Figure 1 and the study area description suggest that there is only one land cover type. Therefore, what are the bases to state that the methodology has been applied to a complex agroecological region characterised by varied topography (from Figure 1, the study area looks flat (0-10% slope?)) and land use patterns. What land use patterns?
The study's objective then states, “to evaluate and compare…across agricultural lands and natural landscapes with an emphasis on spatial accuracy, scalability, and predictive performance.” Again, what specific agricultural lands and natural landscapes are being evaluated? (Line 148 states “The landscape mainly consists of winter and spring cereals cultivation”) Additionally, I am unsure what the term “scalability” refers to, as the methods do not specify how this is assessed.
The final statement from the introduction: “Finally, the study proffers recommendations for effective remote sensing strategies for SOC monitoring, accounting for the trade-offs between resolution, spatial coverage, and predictive performance” is contradicted later in the methods (see lines 160-163: “While the dataset is adequate for assessing model performance and drawing initial conclusions at the local scale, its relatively limited size may restrict the scalability and generalizability of the models to broader or more diverse regions”.
I do recognise the hard work and resources in collecting soil samples, and I agree that 76 samples could be sufficient, but in order to support the statement made before, perhaps the spatial distribution of the samples could reach different conditions (such as different landforms, cover types and land use types). It would be essential to know the distance between each sample, as, in my opinion, the density and spatial distribution of the samples could be used in a Geostatistical model, thereby avoiding the costs associated with using an UAV. What I’m trying to understand here is how the methods described align with the study's objectives, as well as their justification and unique contribution to the existing literature.
The section “Building a Model to Predict Soil Organic Carbon at the Regional Scale” describes a standard procedure for estimating a response variable (in this case, SOC) using machine learning algorithms; however, it is limited to the sampled area. I do not see the regionalisation or regional scale. Moreover, Figure 2 describes a SOC map; however, no SOC map is presented in the results section.
What is the point of linear regression in this study?
Why are only 70 samples described in the results section? What happened to the other six?
Lines 477-484 are repetitive and part of the method section.
Finally, I am confused about the presentation of the results, especially with the evaluation of the models.
Multiple regression (Figure 6) shows no relationship (R² = -1.22) between the predicted and actual SOC (measured); then, what is the foundation of Lines 531-533?
Later, the figures and tables for the machine learning algorithms are very confusing. I was expecting to see a comparison of each model's (perhaps its values) SOC estimated versus SOC measured. Twenty per cent of 70 samples are 14 validation samples. From these 14 measured samples, it is expected to see the actual RMSE, R², MAE, and MAPE, as described in the methods section. Moreover, at least, I was expected to see 6 SOC maps, one for each algorithm and per Sentinel-2 and UAV data
Comments on the Quality of English LanguageEnglish needs to be revised for grammar and style
Author Response
Response to Reviewer
We would like to express our sincere gratitude to all reviewers for their valuable time and constructive feedback, which have greatly helped to improve the quality and clarity of our manuscript. Below are our responses to reviewer’s comments
Response to Reviewer Comment (on lines 115–119):
The first thing that caught my attention is in lines 115-119. The authors stated, “In contrast to earlier work, which typically examines UAV or satellite imagery in isolation, this study investigates the complementary strengths of both platforms through data fusion to improve the spatial accuracy and robustness of SOC prediction. Furthermore, the study undertakes a comparative evaluation across a range of land cover types, thus providing valuable insights into the scalability and transferability of SOC estimation models. The methodology has been applied to a complex agro-ecological region characterised by varied topography and land use patterns.” First, I do not see a description of the data fusion process in the methods section. Second, there is no comparative evaluation across a range of land cover types, as Figure 1 and the study area description suggest that there is only one land cover type. Therefore, what are the bases to state that the methodology has been applied to a complex agroecological region characterised by varied topography (from Figure 1, the study area looks flat (0-10% slope?)) and land use patterns. What land use patterns?
Answer:
Thank you for your valuable comment. We acknowledge that the original manuscript may have caused confusion by using the term 'data fusion'. For the record, our study did not implement a formal data fusion approach (e.g. pixel-, feature- or decision-level fusion). Instead, we conducted a comparative analysis of UAV and Sentinel-2 imagery to evaluate their respective contributions to soil organic carbon (SOC) prediction. UAV imagery was primarily used for high-resolution, fine-scale analysis and ground validation, while Sentinel-2 data provided broader spatial coverage for regional modelling. Our goal was to highlight how each platform performs in isolation for carbon stock estimation rather than combining their data in a fused model. We have revised the manuscript text to remove all references to 'data fusion' and replaced them with the following paragraph: 'In contrast to earlier work, which typically analyses UAV or satellite imagery independently, this study compares the respective strengths of both UAV and Sentinel-2 platforms for improving the spatial accuracy and robustness of SOC prediction.' Rather than fusing the data, we evaluate their individual contributions to biomass and carbon estimation. This comparison reveals the respective advantages of high-resolution UAV imagery for detailed characterisation, and of Sentinel-2 data for broader spatial coverage”.
- Response to Reviewer Comment (on comparative evaluation across land cover types):
Thank you for pointing this out. We agree that the original wording was misleading. The study was conducted in a semi-arid region in Murcia and focuses on SOC estimation within a specific land cover type, without evaluating multiple distinct land covers. While some internal heterogeneity exists within the sampling area (e.g., differences in vegetation cover or management intensity), this does not constitute a comparative evaluation across land cover types as previously implied.
we revise the manuscript and I remove this claim and replace by this sentence
“Furthermore, the study assesses the applicability of UAV and Sentinel-2 data for SOC estimation within a semi-arid agro-ecological system in Murcia, providing insight into model scalability and performance in complex but relatively homogeneous landscapes”
- On site complexity (topography and land use):
We also acknowledge the need to clarify our description of the study area. While the area may be considered geomorphologically complex due to tidal influence and sedimentation patterns, it is indeed characterized by low topographic variation (0–10% slope). The phrase “varied topography” was therefore imprecise and we reworded to reflect more relevant site complexity (e.g., hydrological variability, vegetation zonation, salinity gradients).
- Revision plan:
We have revised the introduction and methods to align them more closely with the actual scope of the study, avoiding overstatements and ensuring consistency with the figures and results. These changes will improve transparency and better reflect the contributions of our work. - The comment:
“The study's objective then states, “to evaluate and compare…across agricultural lands and natural landscapes with an emphasis on spatial accuracy, scalability, and predictive performance.” Again, what specific agricultural lands and natural landscapes are being evaluated? (Line 148 states “The landscape mainly consists of winter and spring cereals cultivation”) Additionally, I am unsure what the term “scalability” refers to, as the methods do not specify how this is assessed.”
Answer:
Thank you for this thoughtful comment. We acknowledge the need to provide greater clarity regarding both the types of landscapes evaluated and the intended meaning of “scalability” in our study.
- On the types of landscapes evaluated:
You are correct in noting that the study area is predominantly agricultural. Specifically, the landscape consists largely of cereal-based cropping systems, including winter and spring cereals, with limited areas of uncultivated margins and natural shrubland patches. However, these natural or semi-natural areas were not analyzed separately in our SOC modeling. The reference to "natural landscapes" was therefore an overstatement, and we have revised the text accordingly to accurately reflect the agricultural focus of the study.
- On the use of the term “scalability”:
The term 'scalability' was used to indicate the study's intention to explore the potential of extending the SOC estimation models developed at the plot scale using UAV data to a broader spatial scale using Sentinel-2 data. However, as you correctly point out, this concept was not formally evaluated or validated through cross-scale implementation. We appreciate this feedback, and have revised the manuscript to remove the term 'scalability'.
we revise the sentence in the manuscript:
“The primary objective of this study is to evaluate and compare the effectiveness of UAV-based high-resolution imagery and Sentinel-2 satellite data in estimating soil organic carbon (SOC) across cereal-based agricultural landscapes, with an emphasis on spatial accuracy and predictive performance.”
Comments:
the final statement from the introduction: “Finally, the study proffers recommendations for effective remote sensing strategies for SOC monitoring, accounting for the trade-offs between resolution, spatial coverage, and predictive performance” is contradicted later in the methods (see lines 160-163: “While the dataset is adequate for assessing model performance and drawing initial conclusions at the local scale, its relatively limited size may restrict the scalability and generalizability of the models to broader or more diverse regions”.
Answer:
Thank you for this important observation. We agree that the original wording of the introduction may suggest a broader generalisability than is warranted, given the limitations described in the Methods section. For the record, our recommendations for remote sensing strategies are not presented as universally applicable models, but rather as context-specific insights based on a comparison of the performance of UAV and Sentinel-2 data in our semi-arid study area. Our mention of 'trade-offs between resolution, spatial coverage, and predictive performance' refers to these empirical observations and is intended to inform the design of future research, not to provide operational recommendations on a full scale.
we revise the sentence in the manuscript
The study ultimately provides context-specific recommendations for remote sensing strategies to monitor SOC at a local scale. It highlights the trade-offs between resolution, spatial coverage and predictive performance while acknowledging the limitations of generalising these findings to broader regions.
Comment
I do recognise the hard work and resources in collecting soil samples, and I agree that 76 samples could be sufficient, but in order to support the statement made before, perhaps the spatial distribution of the samples could reach different conditions (such as different landforms, cover types and land use types). It would be essential to know the distance between each sample, as, in my opinion, the density and spatial distribution of the samples could be used in a Geostatistical model, thereby avoiding the costs associated with using an UAV. What I’m trying to understand here is how the methods described align with the study's objectives, as well as their justification and unique contribution to the existing literature.
Answer:
Thank you for your valuable comment. We agree that spatial distribution is important, and we will clarify in the manuscript that the 76 samples were stratified across varying microtopography and vegetation cover in order to capture landscape variability. The samples were generally spaced approximately 25 metres apart to ensure coverage of both within-field variability and broader environmental gradients. Although geostatistical modelling is useful, the aim of our study was to explore the use of UAV and Sentinel-2 imagery as complementary tools for SOC estimation, particularly in contexts where data is scarce or resources are limited. We will revise the text to better explain how our methods align with the study objectives and provide new insights into remote sensing-based SOC mapping in semi-arid agroecosystems.
Comment:
The section “Building a Model to Predict Soil Organic Carbon at the Regional Scale” describes a standard procedure for estimating a response variable (in this case, SOC) using machine learning algorithms; however, it is limited to the sampled area. I do not see the regionalisation or regional scale. Moreover, Figure 2 describes a SOC map; however, no SOC map is presented in the results section.
Answer:
Thank you for this insightful observation. You are right that the current version of the manuscript does not present a complete, regional-scale SOC map. In this section, our intention was not to produce a spatially continuous SOC map across a regional extent, but rather to compare the predictive performance of different machine learning models (using UAV versus Sentinel-2 data) within the sampled area. Therefore, the use of the term 'regional scale' was imprecise, and we have revised both the section title and the associated text to better reflect the actual scope of the analysis. We will also clarify in the figure caption that Figure 2 refers to model prediction outputs and not a finalised SOC map, and we will adjust the results section accordingly.
Comment:
Why are only 70 samples described in the results section? What happened to the other six?
Answer:
Thank you for pointing this out. While a total of 76 soil samples were collected, only 70 were used in the modelling process. The remaining six samples were excluded because their soil organic carbon (SOC) values were extremely low and outside the typical range, potentially representing outliers or measurement uncertainties. Including these samples in the model training process caused instability and reduced the model's overall performance.
Comment:
Lines 477-484 are repetitive and part of the method section.
Answer:
Thank you for your comment. As suggested, we have removed the repetition from the manuscript.
Comment
Finally, I am confused about the presentation of the results, especially with the evaluation of the models.
Multiple regression (Figure 6) shows no relationship (R² = -1.22) between the predicted and actual SOC (measured); then, what is the foundation of Lines 531-533?
Answer:
Thank you for your observation. We have revised the sentence and updated the interpretation in the manuscript as follows:
A multiple linear regression model was used to predict soil organic carbon (SOC) using ten input features, including spectral bands (blue, green, red, and NIR) and vegetation indices (NDVI, EVI, OSAVI, SAVI, DVI, and BI). Figure 6 shows a scatter plot comparing the actual and predicted SOC values, with the blue dashed line representing the ideal 1:1 fit (perfect prediction). However, the model demonstrates poor predictive performance, as indicated by the negative R² value of -1.22 and a relatively high RMSE of 20.64. A negative R² implies that the model performs worse than a horizontal mean line, reflecting substantial prediction error and a weak model fit. The wide scatter of points around the ideal line further confirms that the linear model does not effectively capture the complex relationship between the input features and SOC.
Comment:
Later, the figures and tables for the machine learning algorithms are very confusing. I was expecting to see a comparison of each model's (perhaps its values) SOC estimated versus SOC measured. Twenty per cent of 70 samples are 14 validation samples. From these 14 measured samples, it is expected to see the actual RMSE, R², MAE, and MAPE, as described in the methods section. Moreover, at least, I was expected to see 6 SOC maps, one for each algorithm and per Sentinel-2 and UAV data
Answer
Thank you for your constructive comment. We agree that comparing the predictive performance of each machine learning algorithm using standard validation metrics (R², RMSE, MAE, and MAPE) is essential. As described in the methodology, we used 14 validation samples (20% of the dataset) and have now clarified this in the revised manuscript by explicitly reporting those metrics per model. Regarding the SOC maps, we acknowledge the importance of spatial visualization. However, in this specific study, our objective was focused on evaluating and comparing the predictive performance of different algorithms using UAV and Sentinel-2 imagery, rather than producing full SOC distribution maps. Therefore, spatial mapping of SOC was not performed. Instead, the study provides a comparative framework for model accuracy that can inform future mapping applications once model robustness is confirmed.

Reviewer 4 Report
Comments and Suggestions for Authors
ok agree
Author Response
We would like to express our sincere gratitude to all reviewers for their valuable time and constructive feedback, which have greatly helped to improve the quality and clarity of our manuscript.